# i-PHAOS: An Overview with an Open-Source Collaborative Database on Miniaturized Integrated Spectrometers

**DOI:** 10.3390/s24206715

**Published:** 2024-10-18

**Authors:** Carla Maria Coppola, Martino De Carlo, Francesco De Leonardis, Vittorio M. N. Passaro

**Affiliations:** Photonics Research Group, Dipartimento di Ingegneria Elettrica e dell’Informazione, Politecnico di Bari, Via E. Orabona, 4, 70126 Bari, Italy; martino.decarlo@poliba.it (M.D.C.); francesco.deleonardis@poliba.it (F.D.L.); vittorio.passaro@poliba.it (V.M.N.P.)

**Keywords:** integrated spectrometers, spectral analysis, miniaturization, integration technology, database

## Abstract

On-chip spectrometers are increasingly becoming tools that might help in everyday life needs. The possibility offered by several available integration technologies and materials to be used to miniaturize spectrometers has led to a plethora of very different devices, that in principle can be compared according to their metrics. Having access to a reference database can help in selecting the best-performing on-chip spectrometers and being up to date in terms of standards and developments. In this paper, an overview of the most relevant publications available in the literature on miniaturized spectrometers is reported and a database is provided as an open-source project to which researchers can have access and participate in order to improve the share of knowledge in the interested scientific community.

## 1. Introduction

Important information concerning an object is carried by its spectrum; the temperature, as well as the chemical composition and the relative speed with respect to the observer, can be derived by an accurate analysis of the absorption and/or emission spectrum of a body. An optical spectrometer is an instrument used to investigate the wavelength-dependent properties of light; it is able to solve the frequency components of light, measuring the intensity of power associated to each frequency component. The use of tools to analyze spectrum has increasingly shown its power, demanding spectrometers to decrease their sizes and be implemented in everyday life devices such as smartphones. The process of miniaturization is not trivial, and it copes with multiple challenges. Indeed, on one hand, one should find the best integration technology to have a compact and low footprint device. On the other hand, the resolution should be as high as possible to guarantee a meaningful spectral analysis of a sample. The miniaturization of optical spectrometers is attracting the interest of the scientific community not only for academic but also for commercial reasons. Indeed, the future market for miniaturized spectrometers is foreseen to be worth 900 million dollars [1].

The increasing demand for portable spectral analysis devices is pushing the miniaturization of these devices towards the centimeter and millimeter scale. The aim of realizing lab-on-chips or smartphone-based sensing devices is driven by a wide range of applications including environmental monitoring, biomedical diagnoses, food and beverage analysis, remote sensing, chemical analysis, astronomy, security and defense. The first review paper discussing miniaturized on-chip spectrometers is dated 2004 [2], where the work by Goldman et al. published in 1990 [3] is reported as the first example of a miniaturized optical spectrometer for chemical analysis based on planar waveguides and grating couplers. Since then, many works have been published increasing the knowledge and the development of new models, architectures and technologies; this led to the need to classify and categorize integrated spectrometers. Recently, detailed reviews have then been published, focused on the description of the basic working principles and integration technologies used to fabricate micro- and nanospectrometers [4,5,6].

The ability to resolve very-close-by optical wavelengths often requires long optical paths, leading to a trade-off between size and resolution, thus explaining the challenge in the miniaturization of high-performance spectrometers. Apart from size and resolution, there are other important parameters to be optimized in the design of a high-performance integrated spectrometer, namely spectral range, bandwidth-to-resolution ratio, dynamic range and measuring speed. A plethora of data concerning the main metrics of integrated spectrometers can already be found in the scientific literature. However, a useful tool to collect and compare data has not yet been published. This work is motivated by the urge to organize and easily compare the values of the several relevant metrics in a referenced and updatable database. Together with the collected metrics, processing tools to analyze the data are also provided in terms of Python scripts that can be customized to obtain the desired plots. Thus, differently from previous published reviews on integrated spectrometers, this work contains a robust framework to allow for an easier comparison among metrics and features (like materials and technologies implemented), defining miniaturized spectrometers and photodetectors.

The paper is organized as follows. In the first section, the main metrics that describe a spectrometer are summarized. In the second section, an overview of the classification of miniaturized spectrometers according to their basic working principles is provided, while, in the third section, on-chip spectrometers are characterized according to the materials used to fabricate them. An insight into the mathematical models is also reported both in the second and third sections. In the fourth section, an open-source project is suggested, that collects a database including the fundamental metrics of integrated spectrometers and a Python script to analyze the database entries. The database will be available on GitHub and periodically updated by the community. Conclusions are drawn at the end.

## 2. Spectrometers: Definitions and Performance Metrics

In order to classify and compare spectrometers, it is worth summarizing the main measurable physical quantities typically used to determine their performances and features. In the following. they will be listed, grouped according to the physical measure they relate to and then briefly described.

Spectral range: this represents how large the window of detectable wavelengths is; it is provided as the range between the maximum and minimum distinguishable wavelengths.

Target peak or central operating wavelength: this represents the central wavelength of the spectral range.

Spectral resolution: this gives a measure of the ability to distinguish between close-by wavelengths; it is usually provided as the difference between the two nearest distinguishable input wavelengths.

Bandwidth-to-resolution ratio: this gives the number of distinguishable wavelength channels within the spectral range. It is strictly related to spectral range and spectral resolution.

Dynamic range: this represents the range of input power that can be distinguished by the spectrometer. It is usually given as the ratio between the maximum and minimum input power.

Footprint: this is an important measure of the area occupation of the spectrometer on the chip. Usually, the footprint of an integrated chip is proportional to the cost of the device fabrication.

Measuring speed: this provides a useful way to evaluate the total time required for a full reconstruction of the input spectrum. It can be also given as the maximum rate of spectrum acquisitions.

Together with the above-mentioned measurables, spectrometers can also be described according to other quantities, e.g., their scalability, i.e., the intrinsic possibility to miniaturize them, given their architecture and the fabrication technology. While the miniaturization process makes the field of possible applications of spectrometers wider, it compromises the spectral resolution and the bandwidth [7].

There are other parameters affecting the quality of spectrometers and related to the photodetectors implemented in the spectrometer architecture. Specifically, these parameters are listed below.

External Quantum Efficiency EQE: this is defined as the ratio between the charge carriers (electrons or electron–hole pairs) generated by absorbing photons.

Internal Quantum Efficiency IQE η: this represents the ratio between the number of absorbed photons and emitted electrons (or recombined electron–hole pairs), n${\,}_{carriers}$: (1)$$\eta =\frac{n_{photons}}{n_{carriers}}.$$

Responsivity R [AW${\,}^{-1}$]: this is a wavelength-dependent quantity, defined as the ratio between the output current, $i_{output}$, or voltage, $V_{output}$, produced and the input optical power: (2)$$R=\frac{i_{output}}{P_{input}}\left[{\mathrm{AW}}^{-1}\right].$$

As a function of the frequency, the responsivity can be expressed as follows: (3)$$R\left(f\right)=\frac{R(0)}{\sqrt{1+{\left(\frac{f}{f_{c}}\right)}^{2}}}$$
where $f_{c}$ represents the so-called cut-off frequency, i.e., the frequency at which the response decreases as 1/*e*, while R(0) represents the responsivity for the null frequency.

Noise Equivalent Power NEP [W Hz${\,}^{-1}$]: this is defined as the input power per square root of the bandwidth resulting in a signal-to-noise ratio SNR = 1: (4)$$NEP=\frac{P_{min}}{\sqrt{\Delta f}}=\frac{N}{R}\left[{\mathrm{WHz}}^{-1/2}\right]$$

Given a specific SNR, using the NEP, it is possible to evaluate the lowest input signal to obtain the desired SNR: (5)$$P_{min}=NEP\sqrt{\Delta f}SNR$$

Detectivity D = NEP${\,}^{-1}$: this represents the inverse of the NEP; hence, the higher the detectivity, the better the photodetector: (6)$$D=\frac{1}{NEP}\left[W^{-1}{\mathrm{Hz}}^{1/2}\right]$$

Specific Detectivity D${\,}^{*}$ [Jones]: this is defined as the inverse of the NEP multiplied by the square root of the detection area A of the photodetector: (7)$$D^{*}=\frac{\sqrt{A}}{NEP}\left[\mathrm{Jones}\right]$$
where 1 Jones = cm·Hz${\,}^{1/2}$W${\,}^{-1}$ [8].

Risetime and falltime [s]: given an optical impulse or a square-pulse signal, they are defined as the time at which the produced photocurrent increases from 10% to 90% or decreases from 90% to 10% of the maximum value achievable with constant illumination of the spectrometer, respectively.

−3 dB bandwidth **f**_−3dB_**:** using the definition of responsivity, the −3 dB bandwidth f_−3dB_ is defined as the frequency at which the output power, $R^{2}$, halves: (8)$$R^{2}\left(f_{-3\mathrm{dB}}\right)=\frac{1}{2}R^{2}\left(0\right)$$

Full width half maximum (FWHM): the wavelength range where the responsivity is higher than half of its maximum value. According to this parameter, a photodetector can be defined as narrowband or broadband. For a deeper analysis, it is suggested to read the book by Liu [9].

## 3. Mechanism-Based Classification of Spectrometers

The main way to classify spectrometers considers their working principles; in this perspective, spectrometers can be classified as wavelength demultiplexing (WDM) and wavelength multiplexing (WM). In an alternative but equivalent way, they are respectively called split-channel and digital spectrometers. The classification is summarized in a diagram in Figure 1.

### 3.1. Wavelength Demultiplexing Spectrometers

These are based on the spatial separation of different wavelength components of light; this is the reason why they are also referred to as split-channel spectrometers. Because the main components for this first kind of spectrometer are dispersive elements (mainly gratings) or filters, wavelength demultiplexing (WdM) spectrometers can be further classified as dispersive or filter-based ones. Integrated dispersive spectrometers are based on similar principles that are commonly implemented in non-miniaturized spectrometers, for example in the Czerny–Turner [10], or Fastie–Ebert [11,12] configuration spectrometers. In WdM spectrometers, different wavelength components of the input spectrum $S_{in}\left(\lambda \right)$ (hereafter λ represents the wavelength in a vacuum) are spatially separated by the split-channel optical components, and they are detected by *N* photodetectors. The detected current Ii at the *i*-th photodetector is related to the input spectrum $S_{in}\left(\lambda \right)$ via the *i*-th photodetector responsivity $R_{in}\left(\lambda \right)$ and the optical transmitted fraction, $C_{i}$, of the input spectrum to the *i*-th photodetector. $I_{i}$ can then be expressed as follows: (9)$$I_{i}=\int R_{i}\left(\lambda \right)C_{i}\left(\lambda \right)S\left(\lambda \right)d\lambda ,\;\;\;\;\;i=1,\ldots ,N$$

It is worth noting that possible losses are accounted for in $R_{i}\left(\lambda \right)$. The full set of output currents can be arranged in an Nx1 vector, *I*. In a split-channel spectrometer, N corresponds to the bandwidth-to-resolution ratio. Moreover, in an ideal case, $C_{i}\left(\lambda \right)$ is proportional to a delta function centered on the *i*-th filtered wavelength.

Typical components of dispersive spectrometers are gratings, like echelle gratings, planar concave gratings and arrayed waveguided gratings (AWGs). In most of the cases, both before and after interacting with the grating, light freely propagates in slab regions called free propagation regions (FPRs); in the case of AWGs, such regions are named star couplers (see Figure 2). Gratings in integrated spectrometers can be used in both transmission and reflection.

The basic principle of dispersive spectrometers is the multi-path interference of light. The optical paths are designed to have different lengths such that light experiences different known phase shifts. The optical path differences are typically obtained by either confining light in one dimension using slab waveguides and diffracting it via gratings with specific geometries (as in the case of echelle gratings and planar concave gratings, PCGs), or by confining light in two dimensions using waveguides (e.g., strip or rib) with different lengths (as in the case of AWGs). For this reason, the first and second kinds of spectrometers can be labeled as 1D- (or slab) and 2D-phase shift driven spectrometers. In the echelle gratings, light is diffracted via saw-like grooves, that might be placed following the circumference of the so-called Rowland circle. This geometrical figure was suggested by Henry A. Rowland in 1883 [13] as an improvement on the shape of gratings used in optics. The advantages of concave gratings mainly lie in their better light focusing and in the reduction in aberration effects [14]. Rowland circles are also used in designing the free propagation regions of AWGs.

It is worth noting that, in the case of 1D-phase shift-driven spectrometers, the region of the slab where light propagates from the input to the grating is also used for light to reach the output photodetectors. This aspect can suggest an optimized way of using the available space on the chip compared to the case of, e.g., AWGs. However, reflection-type AWGs have been suggested to use the same free propagation region for both the input and output of light [15]. Moreover, the freedom in organizing the architecture of the device enables additional flexibility in the layout design of AWG-based on-chip spectrometers.

In [16], SOI AWGs and PCGs were demonstrated to perform better for high- and low-resolution applications, respectively.

The class of filter-based units in wavelength demultiplexing spectrometers is characterized by the usage of non-resonant filters such as cascaded Mach–Zehnder interferometers and resonant filters, like Fabry–Perot interferometers, photonic crystal (PhC) cavities, Bragg gratings and microdonut or ring resonators. Non-resonant filters are also referred to as finite-impulse response (FIR) filters. The working principle of these spectrometers is suggested by their name. Indeed, different optical filters separate the wavelength components of the input light. Such filters are designed to be as selective as possible in the wavelength domain. For this reason, they are labeled as narrowband filters. Differently from dispersive gratings that only implement a spatial separation of the light components, narrowband filters can both work through spatially or temporally separate light. The spatial separation is typically achieved in arrayed narrowband filters, while tunable narrowband spectrometers implement a temporal separation (see Figure 3).

A review on silicon-based optical microspectrometers implementing Fabry–Perot filters has been presented in [17]. Emadi et al. [18] showed the design and fabrication of a high-resolution microspectrometer based on a Fabry–Perot filter with a linearly variable distance between the dielectric mirrors, integrated with a detector array. The resolution of the spectrometer goes from 2.2 nm in the broadband design down to 0.7 nm in the narrowband configuration (spectral ranges 570–740 nm and 610–680 nm, respectively).

PhC cavities can be used as wavelength filters because of their intrinsic periodic structure, as for the case of Bragg gratings. Spectrometers based on PhC cavities could either be constituted of an array of tuned cavities or of a single dynamically tunable cavity. Liapis et al. [19] fabricated a high-quality factor single tunable PhC cavity in SOI technology. Its resolution is reported to be 0.02 nm around 1550 nm. Sharma et al. [20] developed a spectrometer based on TiO${\,}_{2}$ and SiO${\,}_{2}$ alternating-stacked layers. These materials have been chosen because of their low absorption and high refractive index contrast, that in principle makes it possible to have a spectral range in the visible region (400–700 nm).

Concerning integrated microspectrometers employing ring resonators or microdonut resonators as wavelength filters, in 2016, Zheng et al. [21] suggested a high-resolution spectrometer with a tunable micro-ring resonator filter, with a 19 nm spectral range and 0.15 nm resolution. Eventually, Zheng et al. [22] proposed a remarkable architecture combining ring resonators and AWGs, where a 25.4 nm spectral range and a 0.1 nm resolution were achieved in an extremely reduced footprint compared to devices that purely implement only AWGs. Recently, Chen et al. [23] have suggested a spectrometer based on a micro-ring resonator array, having a very compact footprint, an operating spectral range greater than 12 nm and a spectral resolution of 0.17 nm.

As in Zheng et al. [22], it is also possible to find mixed approaches that include both dispersive and filter-based optical components in the same prototyped miniaturized spectrometer. For example, in [24], a biochemical sensing spectrometer implementing both a cascaded micro-ring resonator and an AWG was shown, with a high spectral resolution of 0.42 nm and spectral range of 90 nm. In 2022, Zhang et al. [25] proposed an integrated spectrometer based on a tunable micro-ring resonator and an AWG, with a 70 nm optical spectral range and 0.2 nm resolution. In other designs, several filters of different kinds have been used in the same on-chip spectrometer. For example, Horie et al. [26] adopted a Fabry–Perot filter together with Bragg reflectors while Alshamrani et al. [27] used an add-drop ring resonator as a narrowband filter, combined with a distributed Bragg reflector as a broadband filter.

The bandwidth-to-resolution ratio of arrayed-filter spectrometers corresponds to the number of the adopted filters, while the resolution is limited by the transmission function of each filter.

A description of the differences among the performance metrics of dispersive (AWGs, PCGs) and filter-based (Mach–Zehnder filters) components developed in SOI technology that can be used to realize spectrometers can be found in [28].

### 3.2. Wavelength Multiplexing Spectrometers

As suggested by their name, the wavelength multiplexing spectrometers are based on the principle of wavelength multiplexing. They can be divided into Fourier transform spectrometers (FTSs) and reconstructive spectrometers (or computational spectrometers, CSs, or bandgap dispersion spectrometers), typically implementing Fourier transform interferometers and sets of distinct tunable broadband filters, respectively.

#### 3.2.1. Fourier Transform Spectrometers (FTSs)

FTSs are based on the interference of light. Their main advantages are a large optical throughput and a high signal-to-noise ratio, which allow them to be suitable in detecting even faint signals in dark conditions. The architecture of the FTSs consists of a series of interferometers; the output signal power at the detector is as follows [29]: (10)$$P_{out}\left(l\right)=\frac{1}{2}\int_{0}^{\infty }S\left(\nu \right)\left[1+\cos\left(2\pi \nu l\right)d\nu \right]$$
where ν is the wavenumber in vacuum of the input light and l is the optical path difference between the interferometer arms. By defining ${\bar{P}}_{out}\left(l\right)=2P\left(l\right)-P_{out}\left(0\right)$ (that is called the interferogram) and using the Fourier theorem, ${\bar{P}}_{out}\left(l\right)$ and S(ν) are related as follows: (11)$${\bar{P}}_{out}\left(l\right)=\int_{0}^{\infty }S\left(\nu \right)\left[\cos\left(2\pi \nu l\right)d\nu \right]⟺S\left(\nu \right)=\int_{0}^{\infty }{\bar{P}}_{out}\left(l\right)\left[\cos\left(2\pi \nu l\right)dl\right]$$

In practice, the integration to be performed to obtain the input spectrum is carried out up to a maximum value of optical path difference $l_{0}$. So, the input spectrum can be approximated as follows: (12)$$S\left(\nu \right)\approx \bar{S}\left(\nu \right)=\int_{0}^{l_{0}}{\bar{P}}_{out}\left(l\right)\cos\left(2\pi \nu l\right)dl$$

Since the expression for S(ν) can be rewritten as follows: (13)$$S\left(\nu \right)=\frac{1}{2}\int_{-\infty }^{+\infty }{\bar{P}}_{out}\left(l\right)\exp\left(-2\pi i\nu l\right)dl$$

S(ν) can also be approximated as follows: (14)$$S\left(\nu \right)\approx \bar{S}\left(\nu \right)=\frac{1}{2}\int_{-l_{0}}^{l_{0}}{\bar{P}}_{out}\left(l\right)\exp\left(-2\pi i\nu l\right)dl$$

Limiting the integration domain corresponds to multiplying by a box-car function (equal to 1 between S(ν) and $\mathrm{sinc}\left(l_{0}\nu \right)$). According to the Rayleigh criterion, the resolution is thus obtained as follows: (15)$$\delta \nu =\frac{1}{l_{0}}$$

Consequently, the input spectrum can be approximated at a discrete finite number of wavenumbers (between $\nu_{0}$ and $\nu_{K}$). Moreover, experimentally the interferograms are collected at discrete and finite values of optical path differences equal or lower than $l_{0}$, so the interferogram can be written as follows: (16)$${\bar{P}}_{out}\left(l_{n}\right)=\sum_{k=0}^{K}\bar{S}\left(\nu_{k}\right)\left[\cos\left(2\pi \nu_{k}l_{n}\right)\right],\;\;\;\;\mathrm{for}\;n=0,\ldots ,N$$

Equation (Equation 16) can be solved with different numeric methods (i.e., minimum least squares), in order to obtain ${\bar{S}}_{\nu_{k}}$.

There are several advantages related to FTSs if compared to a wavelength demultiplexing spectrometer. These advantages are usually named after the scientists that underlined them, namely Peter Berners Fellgett, Pierre Jacquinot and Janine Connes. Fellgett’s advantage concerns the possibility of retrieving information from all wavelengths at once: because the scanning time is reduced, this also implies a higher signal-to-noise ratio [30]. Jacquinot’s advantage is also called the throughput advantage, and it is related to the fact that FTSs possess a larger input area compared to the linear apertures usually adopted in dispersive spectrometers, thus allowing for a higher throughput of incident light. Connes’s advantage [31] is connected to FTSs’ higher wavelength accuracy. FTSs can be subcategorized into on-chip static spectrometers and on-chip spectrometers with moving parts. An interesting review on miniaturized FTSs has been recently published [32]. Considering only the static spectrometers, it is possible to have FTSs that are spatially, temporally or spatially-and-temporally modulated. In the first group, it is possible to mention the stationary-wave integrated FTSs (SWIFTSs) and the spatial heterodyne FTSs (SH-FTSs). SWIFTSs can be built using two different arrangements, called the Lippmann and the Gabor (or counter-propagative) configurations, respectively. They are named after Gabriel Lippmann and Dennis Gabor, both awarded with the Nobel Prize in Physics. In both configurations, nanodetectors are periodically placed on the waveguides in order to feel the evanescent field of the waveguides themselves. The Lippmann configuration consists in the injection of light in a waveguide where a mirror is placed at the end, while the Gabor configuration corresponds to the case in which there is light input from both the extremities of the waveguide; in the latter case, light is injected into the waveguide extremities via a Y-junction. When a monochromatic light enters in the SWIFTSs, both in the Lippmann and in the Gabor configurations, stationary waves are formed along the waveguides. However, if white light is injected, different interference patterns will appear. Specifically, in the case of the Lippmann configuration, the so-called Lippmann interferogram is created, characterized by the presence of the black central fringe at the mirror interface and resulting from the superposition of several stationary waves. In the case of the Gabor configuration, the waves interfere in a counter-propagating way, thus producing a Fourier interferogram with the black fringe at the center of the device when the difference in their phase is null. In the Gabor configuration, it is then possible to have information both on the spectrum and on the phase of the interfering electromagnetic waves (Figure 4, see also [33,34]). The first integrated implementation of a SWIFTS has been reported by Cavalier et al. 2011 [35], where superconducting nanowires are used as single-photon detectors along a SiN waveguide. An integrated implementation of this configuration can be found in [36], where the interferograms are under-sampled in the interference region with golden nanorods and obtained by changing the phase of the input light signals using electro-optical modulation.

Temporally modulated spectrometers are also called temporal heterodyne FTSs (TH-FTSs) or active scanning FTSs because of the presence of both passive and active parts on the photonic integrated circuit (hereafter PIC). The active part is performed by means of the thermo-optical or electro-optical effect. In TH-FTSs, the interferogram is obtained as an ensemble of output power collected at different times, corresponding to different applied phase shifts. In Figure 5, an example of SH-FTSs is reported, where the differences in the optical paths are achieved using unbalanced Mach–Zehnder interferometers (MZIs). SH-FTSs typically employ arrays of unbalanced Mach–Zehnder interferometers with linearly increasing differences in the optical path differences. In Heidari et al., 2019, an example is provided using silicon-on-sapphire technology [37]. In this configuration, the interferogram is obtained as the collection at the output at spatially displaced photodetectors.

In Figure 6, a schematic example of the working principle of a TH-FTS is reported. Typically, the phase change ΔΦ can be achieved either via the thermo-optic or the electro-optic effect. In Zheng et al., 2019 [38], an example of a spatially-and-temporally modulated FTS is reported; such an alternative FTS is obtained by cascading a tunable micro-ring resonator with a tunable Mach–Zehnder interferometer. More recently, Xu et al., 2024 [39] demonstrated an integrated spectrometer with a resolution of 250 pm over a 200 nm bandwidth with a configuration implementing both spatial and temporal modulation, referred to as a 2D Fourier transform spectrometer. Li et al., 2024 [40] suggested an inversely designed integrated spectrometer in which the spectrometer can work under two different regimes and it is based on a programmable photonic circuit.

#### 3.2.2. Reconstructive Spectrometers (or Computational Spectrometers) (RSs)

Reconstructive spectrometers are special kinds of spectrometers in which the input spectrum is acquired at all wavelengths at once and it is reconstructed by means of postprocessing algorithms applied to the output optical powers collected by a photodetector matrix. In particular, they are designed such that the input light interacts with special optical components, designed such that their response functions ensure the orthogonality among output vectors.

The reconstruction procedure is based on the theoretically demonstrated finding by Wang and Yu in 2014 [41] that random spectral filters can help in gaining high spectral resolution. This can be achieved together with using advanced signal processing methods, e.g., compressive sensing or others (see, e.g., [7,42,43,44,45] for a detailed description). Based on this finding, the first experimental computational spectrometer has been shown in the seminal work by Bao and Bawendi [46]. Successive works have been conducted by [47,48,49]. RSs can also be called bandgap dispersion spectrometers because the typical basic elements in the optical detection are photonic crystal slabs or nanostructured semiconductors. In general, broadband filters are implemented in RCs as detecting units that separate light components. As in the case of narrowband filters, broadband filters can be also arranged as an array or in a tunable configuration (Figure 7).

Given an unknown input spectrum S(λ) and *N* photodetectors, it is possible to express the measured photocurrent $I_{i}$ at each ith photodetector as follows: (17)$$I_{i}=\int R_{i}F_{i}\left(\lambda \right)S\left(\lambda \right)d\lambda \equiv \int D_{i}\left(\lambda \right)S\left(\lambda \right)d\lambda ,\;\;\;\;\;i=1,\ldots ,N$$
where $R_{i}\left(\lambda \right)$ is the responsivity of the *i*-th photodetector and $F_{i}\left(\lambda \right)$ corresponds to the transmission function of the adopted broadband filter. For convenience, we define their product as $D_{i}\left(\lambda \right)$. Discretizing Equation (Equation 17), the outputs of the photodetectors can then be arranged as an N×1 vector: (18)I=D·S
where *I* is an N×1 vector, *D* is an N×P matrix and *S* is a P×1 vector, *P* being the number of discretized wavelengths. The transmission spectral response matrix *D* is obtained via calibration procedures. Typically, a calibration laser source is scanned over a certain wavelength range and the output intensity *I* is recorded. The transmission matrix *D* relates the spectral domain to the spatial domain in a not-trivial one-to-one way, as it occurs in the case of classical split-channel spectrometers. In reconstructive spectrometers, each row of the matrix *D* contains information on the transmission spectrum at different wavelengths. Eventually, it is possible to recover the unknown spectrum by inverting the transmission matrix in Equation (Equation 18): (19)$$S=D^{-1}\cdot I$$

The matrix inversion is a numerically unstable procedure because of the intrinsic fluctuation connected to the noise [50]. For this reason, it is usually performed together with a non-linear optimization procedure. The most commonly used reconstruction algorithm is based on the so-called l1 norm optimization, in the convex or non-convex version, while other algorithms have been suggested, such as the greedy algorithm and Bayesian method. The main reference papers for these algorithms have been published during the first decade of the 2000s. A comparison between several compressive sampling strategies for integrated spectrometers can in found in [51]. In [52], an interesting framework to compare different compressive algorithms can be found. This open-source project aims at providing a standardized tool to develop and perform image compressive sensing. The updated git repository can be found at the link https://github.com/PSCLab-ASU/OpenICS (accessed on 3 May 2024). This approach can be applied even in the N << P case, i.e., when the number of broadband filters is smaller than the discretization dimension of the input spectrum. Such a condition corresponds to undetermined linear systems often prone to being ill conditioned [51]. To prevent ill-conditioning effects on the spectrum reconstruction, the resolution of Equation (Equation 19) can be performed using reconstructive procedures that implement regularization algorithms. Together with the usage of such algorithms, reconstructive spectroscopy can proceed by using compressed sensing. Compressed algorithms for signal processing are based on the idea of measuring a spectrum using a limited number of measurements and reconstructing it via reconstruction algorithms by using a compressed version of such measurements [53]. Among the suggested reconstructive algorithms, neural networks have also been adopted [54]. Several approaches have been developed as reconstruction techniques, namely the so-called speckle-spectroscopy, filter-array reconstruction spectroscopy and stochastic optical reconstruction spectroscopy (STORS) techniques [45]. RSs have been fabricated by using multimode optical fibers [55], colloidal quantum dots [46], structurally engineered silicon nanowires [56], a single nanowire [57], black phosphorus [58], single-dot perovskite [59], a superconductive nanowire [60] and parallel cascaded micro-ring resonators [61]. It is worth noting that spectrometers like the ones described in [59,60,62,63] could be considered as a subset of RSs in which the spectrometer is externally biased N times in order to fill the responsivity matrix $R_{i}\left(\lambda \right)$ and the spectrum is detected at each bias by a single photodetector (rather than having *N* distinct photodetectors acquiring the input spectrum filtered by *N* broadband filters). Oliver et al. [64,65] showed that it is possible to work both on the reconstructive algorithms and the transmittances in order to improve the resolution of the spectrometer; in particular, the resolution of spectrometers increases with hyper-random transmittances. In 2021, Sharma et al. [20] have developed a photonic crystal-based reconstructive spectrometer with alternating layers of TiO${\,}_{2}$ and SiO${\,}_{2}$. It has been recently suggested [66,67] to use photonic molecules (specifically, in [66], four microdisk photonic atoms are adopted) in order to spectrally discriminate light. The spectral resolution is demonstrated to be ∼8 pm, though keeping the footprint very small (70 × 50 μm${\,}^{2}$). This approach paves the way for further miniaturization scales. The application of reconfigurable photonics has been recently adopted to produce a broadband high resolution (pm) integrated spectrometer, that uses the reconstructive approach to derive the output spectrum [68,69]. An exotic spectrometer has been suggested by Kwak et al. [70] that exploits the optical features of mother of pearl. A recent review on RCs has been published [71].

A comparison of cons and pros for each kind of spectrometer is summarized in Table 1. According to this information, it can be pointed out that the choice of a specific type of spectrometer is influenced by the particular application it is supposed to be used for. The speed of processing the input spectrum, for example, is crucial for the analysis of substances that might be hazardous for health in medical and real-time applications while a higher resolution is required to distinguish spectrum fingerprints in chemical compounds, for example, in environmental and astrophysical applications. The level of integration (hence, the size of the full spectrometer) and the kind of adopted platforms also influence the applications they can be used for. This evidence represents an additional reason to have a referenced database with all this information collected.

## 4. Spectrometer Classification Based on Material Type: Inorganic, Organic and Metamaterials

It is useful to classify miniaturized (micro- and nano-)spectrometers and their photodetecting optical components according to the materials that are adopted to fabricate them. Indeed, an insight into the chemical compounds suggests their efficiencies and their potential usages in specific applications. Moreover, the analysis of the materials is beneficial to understand the basic working principles and, as a consequence, the optimal architecture to be adopted. It is possible to group the spectrometers into inorganic material- and organic material-based devices. Together with these categories, it is worth adding a third class, represented by metamaterial-based miniaturized spectrometers. In the first two classes, the main photodetective materials are semiconductors.

### 4.1. Inorganic Material-Based Spectrometers

The conventional semiconductor used to fabricate integrated spectrometers is silicon, the second more abundant chemical species on Earth. Among other reasons, the usage of silicon for integrated optical circuits is promoted for full compatibility with mature CMOS technologies. There are several technologies in PICs that implement silicon, namely SOI (an acronym that stands for silicon on insulator) and silica on silicon. The high refractive index contrast of silicon (n${\,}_{Si}$ = 3.476) and its oxide SiO${\,}_{2}$ (n${\,}_{{SiO}_{2}}$ = 1.444) at λ = 1550 nm is relatively high; hence, the light can be waveguided with very small cross-sections and reduced bend radii. As a consequence, the footprint associated with such devices is small. Spectrometers fabricated using silicon-based technologies can be found in several papers [15,27,28,72,73,74,75,76,77]. Another IV group semiconductor element that is used in spectrometers is germanium (Ge) [77], whose corresponding technology for integrated circuits is named GOS (germanium on silicon). For a GOS platform, the contrast index is even higher than the SOI technology, the refractive index of germanium being n${\,}_{Ge}$ = 4.2162 for λ = 1550 nm (the highest refractive index in the case of this element is found for infrared radiation). In integrated spectrometers, Si and Ge can be used both for optical passive components and for photodetectors, according to the absorption of the chosen material in a certain operational wavelength region. Specifically, the transparency windows of silicon and germanium are approximately [1.9–12] μm and [1.2–7] μm, respectively; hence, in these regions, they can be used to realize passive optical components. Instead, in the absorption wavelength regions, they can be adopted to realize photodetectors (although the absorption is limited by them being both indirect semiconductors). It is possible to have composite semiconductors, using elements from the III–V, II–VI and IV–IV groups. Additional binary semiconductors that are used in photonic devices and give the name to the corresponding technologies for optical device platforms are indium phosphide (InP) and silicon nitride (Si${\,}_{3}$N${\,}_{4}$). III–V group semiconductors have direct energy gaps: this feature leads to having high absorption probabilities and tunable energy gaps, hence broader spectral ranges and higher sensitivities. Moreover, these platforms can be heterogeneously integrated on chip, as well as monolithically, to have both light sources and photodetectors on the same chip. III–V group materials can be used as gain materials for lasers and LEDs. This aspect makes them appealing in order to achieve lower dimensions and highly dense integrated spectrometers [78]. SiN devices are also compatible with CMOS technologies, and they have the additional advantages of having a very low propagation loss and broadband spectral range, practically not influenced by thermal variations. Several spectrometers have been recently suggested using Si${\,}_{3}$N${\,}_{4}$ [79,80,81,82]. Previously, Nie et al. [83] reported the development of a Fourier transform spectrometer integrated on a silicon nitride platform with an extremely small footprint (0.1 mm${\,}^{2}$) and 6 nm resolution. Several materials have been used to fabricate PCGs and AWGs, such as silica-on-silicon, SOI (silicon-on-insulator), GOS (germanium-on-silicon) and Si${\,}_{3}$N${\,}_{4}$ (silicon nitride) integration technologies [84]. It is worth introducing and distinguishing the concepts of hybrid and heterogeneous integration that can be used to realize PICs. The former refers to the integration process according to which different photonic integrated chips are bonded together to form a single device. Typically, in hybrid integration, the starting chips are made of different materials and/or they have been fabricated using different technologies. Heterogeneous integration, on the contrary, concerns the integration process with which different materials and/or technologies are integrated on the same photonic chip. This integration option has major advantages like being very similar to monolithic integration, having lower fabrication costs than hybrid circuits and allowing for an easier alignment of the optical elements. Because of this last advantage, optical power losses are moderate at the interface between waveguides, although implemented using heterogeneous technologies. An interesting description of the experimental methods used to perform both hybrid and heterogeneous integration in optical devices can be found in [85]. Heterogeneous integration approaches have been adopted in fabricating integrated optical spectrometers [86,87,88].

### 4.2. Organic Material-Based Spectrometers

Nowadays, in order to overcome the high fabrication costs of inorganic semiconductors, organic compounds are suggested as a feasible and interesting alternative. In fact, the usage of some polymers in order to create a planar grating spectrograph dates to several decades ago [89], where Polymethylmethacrylate (PMMA) was adopted in order to create the dispersive structure of an infrared spectrometer. More recently, research has been conducted in the field of polymer based organic photodetectors (OPDs). Narrowband OPDs can be easily implemented to fabricate an integrated filter-based spectrometer. OPDs’ standard design consists in a heterojunction containing compounds that behave as an electron acceptor and some other blend behaving as an electron donor. For this behavior, acceptors and donors resemble the inorganic n- and p-type semiconductors, respectively, hence the name of organic semiconductors (OSs). It is possible to either deposit a donor layer followed by an acceptor layer (in this case a planar bilayer heterojunction is created) or acceptors and donors can be blended together to form bulk heterojunctions (BHJs) (Figure 8). The region where the acceptor and donor are placed represents the photo-active part, typically tens of nanometers thick. As in the case of inorganic semiconductors, this region is put in between the electrodes. In this case, the absorption of the photons occurs through one of the electrodes that is semi-transparent (for example, based on indium tin oxide—ITO). In organic semiconductor materials, the absorption of photons induces the formation of a quasi-particle called an exciton. When an exciton dissociates, free charge carriers are generated, contributing in principle to the photocurrent when they can reach the electrodes. However, the mobility of these charge carriers is limited compared to the inorganic counterparts (up to four orders of magnitude lower) due to the different interaction between electrons and periodic potentials, the presence of impurities or defects in the crystals, and amorphous or crystalline nature of the compounds.

The colour discrimination in such devices can be mainly achieved either using a special kind of effect that can be exploited in organic compounds or via the specific design of the architecture of the spectrometer itself. For the first category of approaches, the mainstream consists in using so-called charge collection narrowing (CCN) [90]: in an optically and electrically thick heterojunction, the transit times for the carriers depend on the wavelengths of the incoming photons. For this reason, a narrowband response can be achieved by exploiting the different times. For the second approach, cavity-enhanced OPDs are usually implemented in order to obtain an integrated spectrometer by tuning the cavity length [47,91,92]. Xing et al. [91] suggested a spectrometer built using a transmission cavity-based organic photodetectors with a remarkable specific detectivity of 10${\,}^{14}$ Jones and spectral range between 400 nm and 1000 nm. The materials used for the active blend are DCV5T-Me:C60, BDP-OMe:C60 and QM1:C60 because of their absorption spectra in the blue, green and red wavelength ranges, respectively. Li et al. [93] recently suggested a prototype spectrometer based on special organic semiconductors (namely quinoid-capped OSs) used as photodetectors working between 400 nm and 1250 nm, with a detectivity that is comparable to InGaAs photodetectors (∼10${\,}^{12}$ Jones).

### 4.3. Metamaterial-Based Spectrometers

Metamaterials are defined as structures with artificially produced low-scale patterns that allow some specific optical properties to be achieved. For this reason, metamaterials are also referred to as reconfigurable materials, having physical properties that can be tuned according to their shapes and sizes. Specifically, metamaterials can be designed to be used in photonics applications [94]. In the case of photodetectors, metamaterials give the possibility of having a good spectral tunability [95]. Moving to the quantum size domain (<10 nm), materials show electrical and optical properties that depend on their size. In particular, for 0D nanostructures like quantum dots, the energy gap increases as the size of the nanoparticles decreases. For 1D metamaterials like nanowires and nanotubes, the tunability of the ranges of the physical hallmarks is achieved by changing the radius of the 1D nanostructures, while for 2D materials the thickness of the layers determines their peculiar optoelectronic properties [96,97]. A very interesting review on the nanomaterials adopted to miniaturize spectrometers has been recently published [6]. The metamaterials typically used as optical integrated components for spectrometers are mainly inorganic, with some of them compatible with complementary metal oxide semiconductor (CMOS) technologies. This feature makes them promising materials to realize integrated on-chip spectrometers. However, organic quantum dots have also been explored as photodetectors in microspectrometers [98]. Among 0D metamaterials, quantum dots (QDs, both organic and inorganic [98]), colloidal quantum dots (CQDs) and perovskite quantum dots are used as optical components in different materials and arrangements [99,100,101]. QDs are also used as filters, as in [102], where an array of PbS and PbSe QDs have been used as an optical filter in a reconstructive spectrometer because of their spectrum tunability. Indeed, in this case, the band gaps depend on the size, the composition and the ligand of the QDs. Wen et al. [103] designed microspectrometers based on HgSe CQDs and distributed Bragg reflectors, working as a resonant cavity. Among 2D metamaterials, hexagonal boron nitride (hBN) and transition group metal sulfides are used because of their very good optical detection properties. In particular, a black phosphorus-based spectrometer has been fabricated, showing a spectral range of 9 μm with a spectral resolution of 90 nm [58]. The bandgap of black phosphorus is intimately related to the thickness of the sample, hence making the spectral range vary according to a macroscopic quantity. A spectrometer based on a MoS${\,}_{2}$/WSe${\,}_{2}$ heterojunction in hBN stacked layers has been suggested [62]. Graphene is also adopted [104,105] in fabricating optical spectrometers. A comprehensive list of 2D materials beyond graphene used in optoelectronics can be found in [106]. Both dispersive gratings and narrowband filters can be produced by using anisotropic metamaterials [107].

## 5. Open-Source Spectrometer Database

So far, an overview of the materials and integration technologies has been presented to show on one hand the richness and variety of data existing in the scientific literature, on the other hand to suggest the need for a referenced framework in which important information could be easily found. The information has been organized according to the diagram and keywords shown in Figure 9. All the data available in the literature and concerning integrated and micro-/nanospectrometers have been usefully collected in the database i-PHAOS: integrated PHotodetectors And Optical Spectrometers, in which the metrics for each spectrometer are easily accessible for comparison. In fact, given the plethora of available data, we suggest here a referenced database available on GitHub at the link https://github.com/carlamariacoppola/iPHAOS (released under the GNU v3.0 license). The database is provided as an Excel file composed of two sheets named “spectrometers” and “photodetectors for spectroscopy”. A third sheet is provided with the drop-on entries included in the previous sheets. The entries for the references are numbered according to the paper bibliography, while each reference is explicitly mentioned and hyperlinked to the corresponding doi page on the GitHub version. The data included in the sheet named “spectrometers” are as follows:1.Type of spectrometer: it is possible to choose the entry as dispersive (gratings) spectrometer, filter-based spectrometer, Fourier transform spectrometer or reconstructive (computational) spectrometer.2.Materials/integrated platform.3.Spectral range: this is sub-categorized as spectral range (min value), spectral range (max value), bandwidth and target peak. All data are provided in nm.4.Spectral resolution [nm].5.Bandwidth-to-resolution ratio.6.Dynamic range.7.Footprint, area [mm${\,}^{2}$].8.Footprint, volume [mm${\,}^{3}$].9.Measuring speed [s].10.CMOS compatibility.11.Year.12.Ref.

Similarly to the best available review table available in the literature on photodetectors by Wang et al. 2022 [92], the sheet named “photodetectors for spectroscopy” is organized according to the following metrics:1.Material.2.Spectral range: this is sub-categorized as spectral range (min value), spectral range (max value) and target peak. All data are provided in nm.3.Spectral resolution [nm].4.FWHM [nm].5.D* [Jones].6.EQE.7.R [A W${\,}^{-1}$].8.Bias [V].9.LDR [dB].10.f${\,}_{-3\mathrm{dB}}$ [kHz].11.Footprint, area [mm${\,}^{2}$].12.Footprint, volume [mm${\,}^{3}$].13.CMOS compatible.14.Year.15.Ref.
Figure 9Materials and integration technologies implemented in miniaturized spectrometers: an overview.
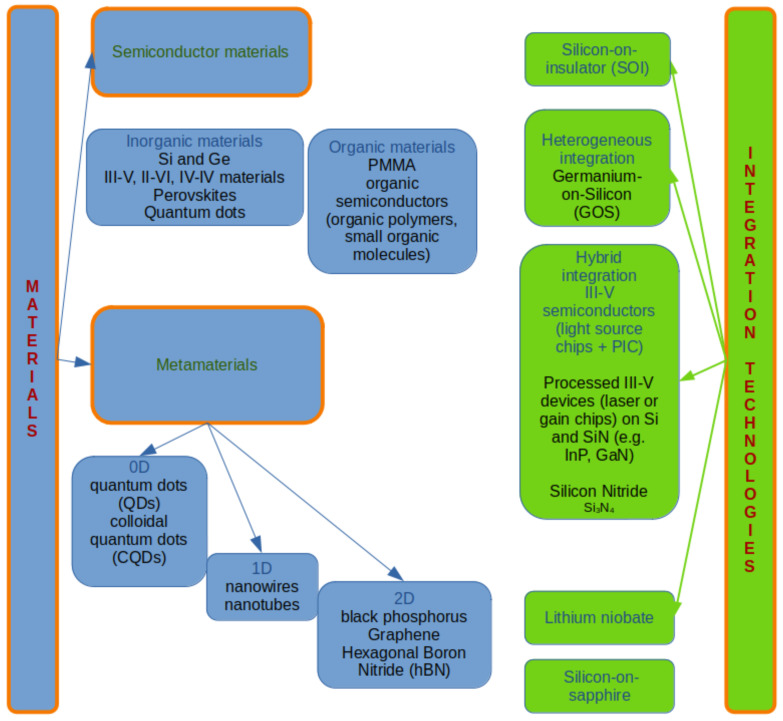


The database available on the GitHub repository can be analyzed by means of a Python 3.10.12 script such as the one provided in the same repository with the filename spectrometers.py. The source file is commented to ease the reader in customizing the queries to the database according to their needs. In this script, the database is imported as a dataframe using the Python library Panda, that is also used to perform the analysis of the entries. In the repository, some examples of plots are provided and produced by using the matplotlib library. We report here some examples of plots that can be produced to quickly perform a comparison among metrics of interest. In Figure 10, the case of the bandwidth in nm for each reference is plotted while in Figure 11 the spectral resolution in nm is reported.

In Figure 12, the case of a 3D scatter plot of the spectral resolution and the bandwidth-to-resolution ratio for each reference is reported. The multidimensional plots are provided such that, whenever an entry for a specific metric (row) of the database is not available, the corresponding reference is dropped from the dataframe by using the method “dropna”. The project is open source and scientists are invited to include their data to increase the stored information and keep track of the changes and improvements in the field. All the papers referenced in the database (version tagged as v0 in the github repository) and not yet cited in this manuscript can be found in references [15,108,109,110,111,112,113,114,115,116,117,118,119,120,121,122,123,124,125,126,127,128,129,130,131,132,133,134,135,136,137,138,139,140,141,142,143,144,145,146,147,148,149,150,151,152,153,154,155,156,157,158,159,160,161].

## 6. Conclusions

Optical spectrometers are important and widely used devices for several applications, including research fields as biomedicine, astronomy, chemistry, physics, communication, etc. The high demand for portable and miniaturized spectrometers is pushing the recent research towards their on-chip integration, down to the micro- and nano-scale. The challenge of this miniaturization process lies in the intrinsic trade-off between the size of the spectrometer and its performances (i.e., spectral resolution, spectral range, measuring speed). One of the exciting possible applications of the integration of spectrometers is the fabrication of devices able to perform the full spectral analysis of analytes to be characterized (e.g., [76]). In this paper, an overview of the main approaches that have been recently followed in the design and development of an on-chip integrated spectrometer has been presented. At the current state of the art, the latest works on miniaturized spectrometers show integrated circuits exhibiting footprints that are fractions of mm${\,}^{2}$ and a bandwidth-to-resolution ratio on the order of hundreds, in some cases thousands. Several technologies and materials are implemented, going from the heavily used silicon (e.g., in the SOI, SOS and SiN platforms) to metamaterials of different dimensionality. Because of the availability of numerous modeled and fabricated devices, a devoted database of the main metrics has been created with the aim of collecting the existing information and processing it. This project has been meant to be open source and collaborative meaning that contributors are invited to send entries to be added to the database via a merge request or email and periodic updates of the database will be released. The database is provided as a tool for people working in the field of research and development of on-chip spectrometers; hence, scientists are invited to update the database on the GitHub repository.

## Figures and Tables

**Figure 1 sensors-24-06715-f001:**
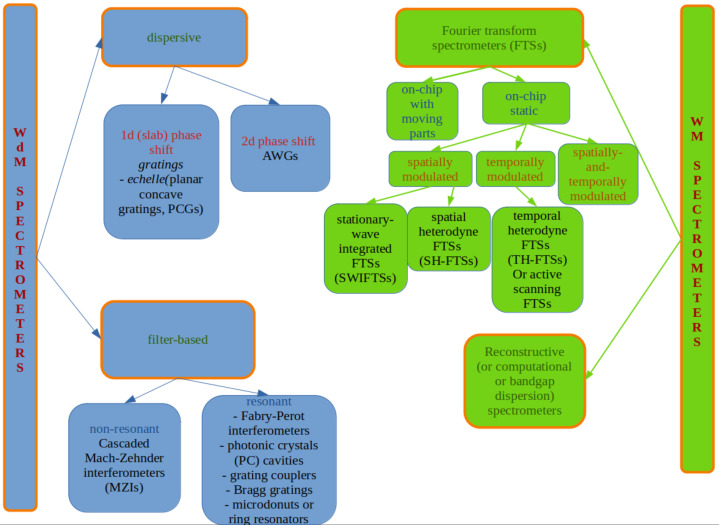
Classification of on-chip spectrometers according to their basic working principles.

**Figure 2 sensors-24-06715-f002:**
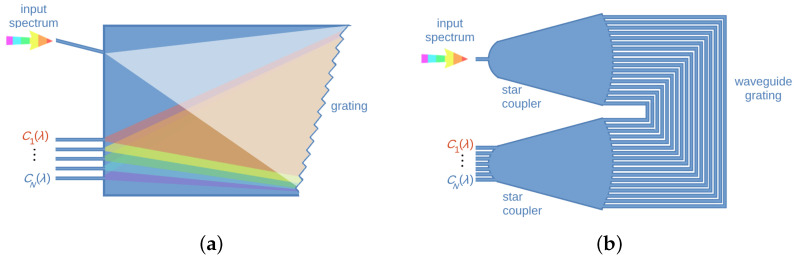
Schematic diagrams describing demultiplexing dispersive spectrometers. (**a**) Grating. (**b**) Arrayed waveguide grating.

**Figure 3 sensors-24-06715-f003:**
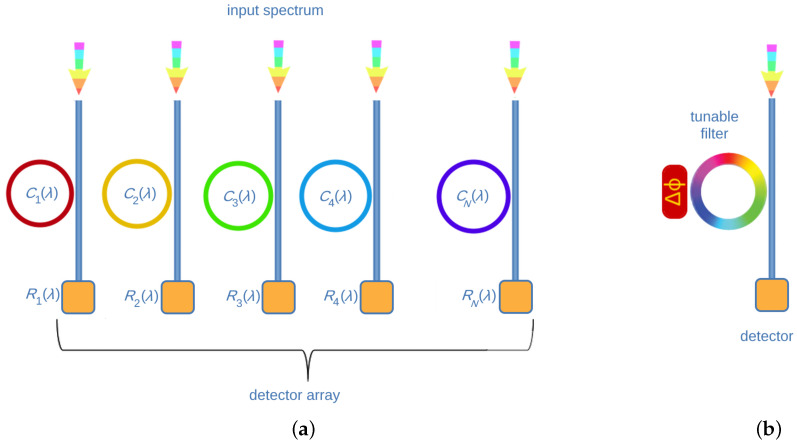
Schematic diagrams describing demultiplexing filter-based spectrometers. (**a**) Arrayed narrowband filters. (**b**) Tunable narrowband filter spectrometer.

**Figure 4 sensors-24-06715-f004:**
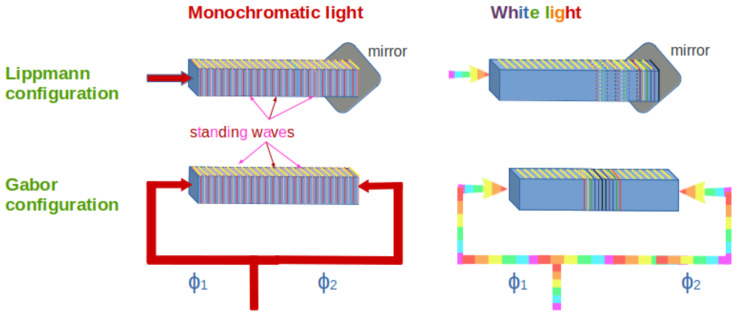
Schematic diagrams describing multiplexing/spatial heterodyne FTSs. Top part: Lippmann configuration, i.e., the light enters in the waveguide and is reflected by a mirror; in the case of monochromatic light, a standing wave forms, while in the case of white input light an interference figure appears close to the mirror, showing a black fringe at that side. Bottom part: Gabor configuration, i.e., the light enters in the waveguide from both sides; if monochromatic, standing wave forms, otherwise the interference figure shows up along the waveguide. The interferogram is centered if the phases of the interfering waves are the same.

**Figure 5 sensors-24-06715-f005:**
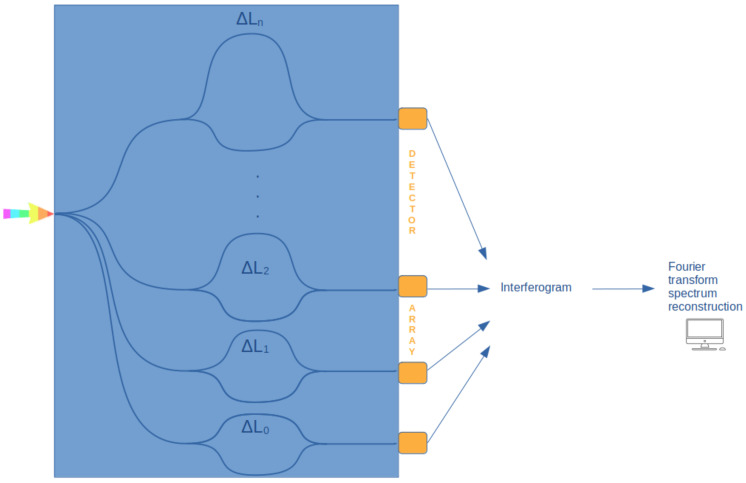
Schematic diagram describing multiplexing spatial heterodyne Fourier transform spectrometers (SH-FTSs).

**Figure 6 sensors-24-06715-f006:**
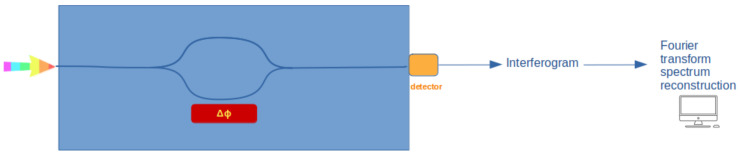
Schematic diagram describing multiplexing temporal heterodyne Fourier transform spectrometers (TH-FTSs) or active spectrometer.

**Figure 7 sensors-24-06715-f007:**
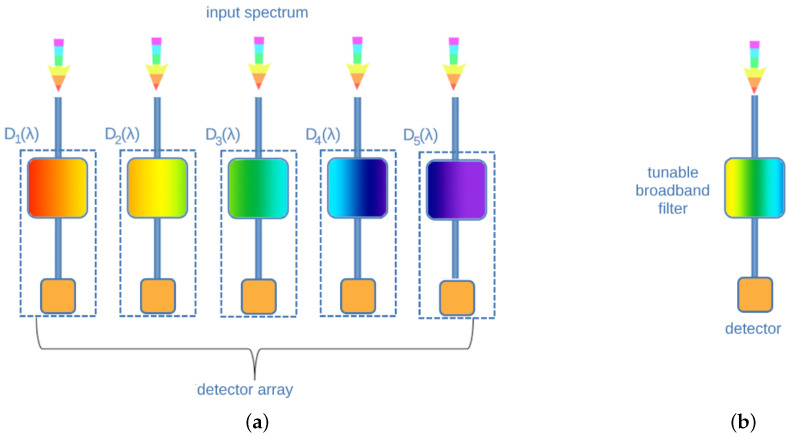
Schematic diagrams describing multiplexing broadband filter-based spectrometers (reconstructive or computational spectrometers). (**a**) Arrayed broadband filters; (**b**) tunable broadband filter spectrometer.

**Figure 8 sensors-24-06715-f008:**
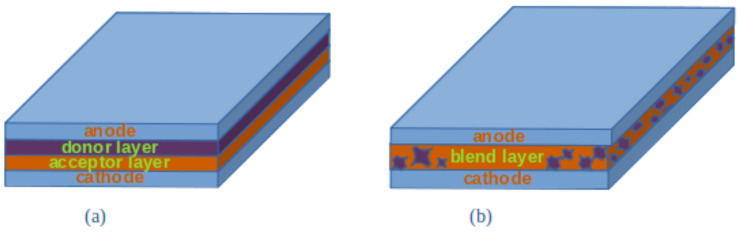
Basic types of organic photodetectors (OPDs). (**a**) Planar bilayer heterojunction configuration. (**b**) Bulk heterojunction (BHJ) configuration.

**Figure 10 sensors-24-06715-f010:**
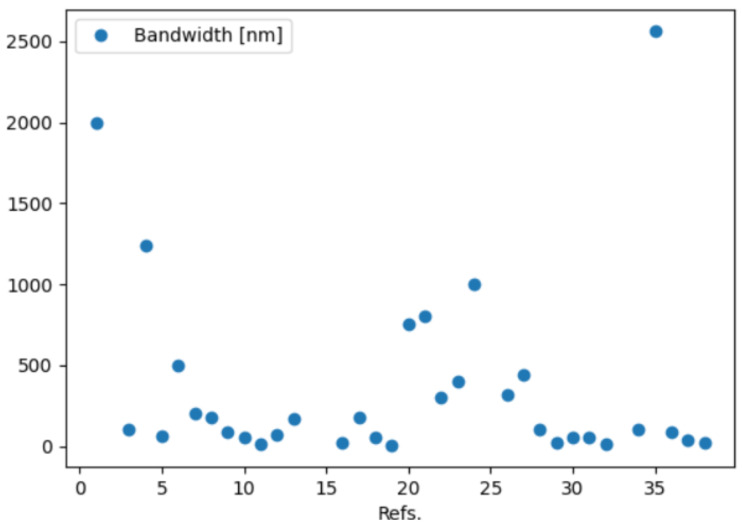
Demonstration of plots that the database analysis can generate: example 1.

**Figure 11 sensors-24-06715-f011:**
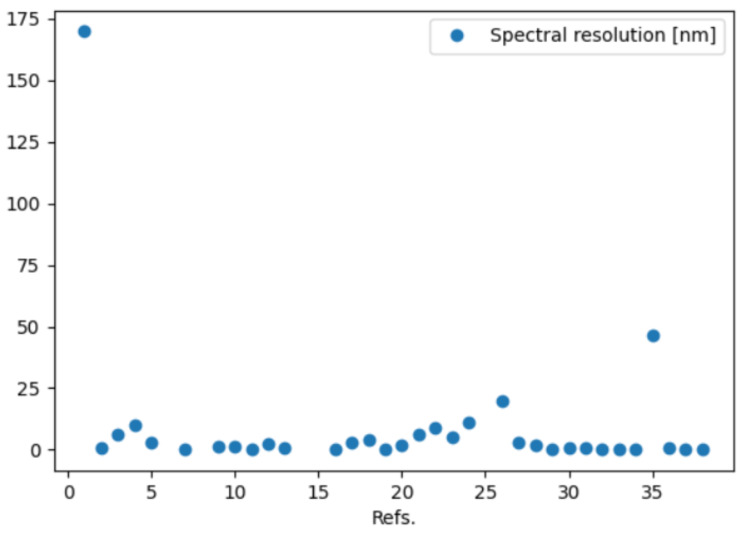
Demonstration of plots that the database analysis can generate: example 2.

**Figure 12 sensors-24-06715-f012:**
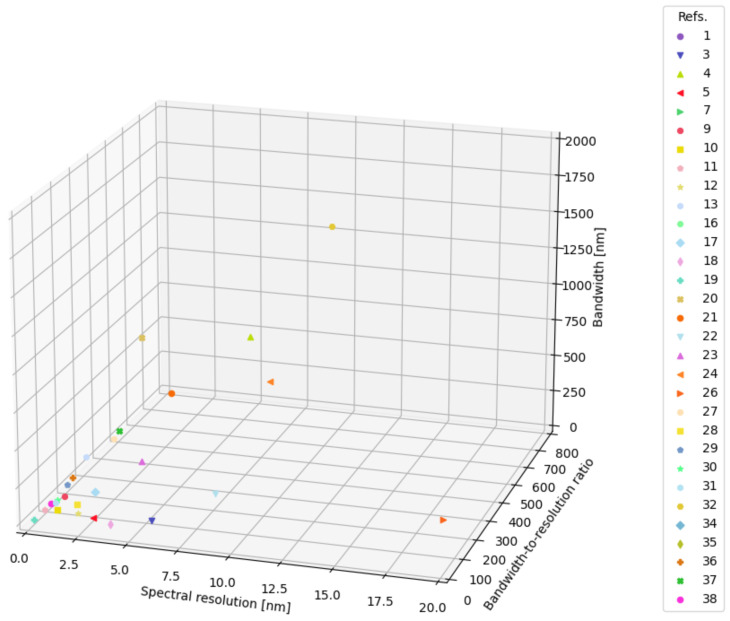
Example of informative plots that can be produced by analyzing the database: 3D scatter plot of the spectral resolution in nm and bandwidth-to-resolution ratio for each reference providing both entries.

**Table 1 sensors-24-06715-t001:** Summary table of pros (+) and cons (-) of the different kinds of integrated spectrometers.

WdM/WM	Spectrometer Type	Subtype	Pros	Cons
WdM	Dispersive	1D (slab) phase shift	+ Simple design + High measuring speed	- Limited spectral range - Low resolution
		2D phase shift	+ High measuring speed	- More complex fabrication - Low resolution - Larger footprint
	Filter-based	Non-resonant	+ Wide spectral range + High resolution + Tunable options	- Lower spectral resolution - Larger footprint - Power hungry
		Resonant	+ High spectral selectivity + Compact and integrable design	- Sensitive to temperature variations - Fabrication complexity
WM	Fourier transform spectrometers	Spatially modulated	+ High spectral resolution + High throughput (Jacquinot advantage)	- Requires precise alignment - Potentially complex readout
		Temporally modulated	+ Very high spectral resolution (depending on delay range)	- Complex for broadband spectrometers - Requires sophisticated processing

## Data Availability

The database and scripts to investigate it can be found via the link https://github.com/carlamariacoppola/iPHAOS (accessed on 25 September 2024).

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
