# Peer review of "i-PHAOS: An Overview with an Open-Source Collaborative Database on Miniaturized Integrated Spectrometers"

_sensors, 2024, doi:10.3390/s24206715_

Round 1
Reviewer 1 Report
Comments and Suggestions for Authors
The paper covers the main types of spectrometers and the latest developments, including from traditional dispersive spectrometers to emerging reconstructive spectrometers. The content is well-organized and logically clear, allowing readers to fully understand the current state of the field. Classifying spectrometers according to their working principles and materials is an innovative perspective, helping to better understand the advantages and disadvantages of different types of spectrometers and their applicable scenarios.
While the manuscript is interesting in general, it is necessary to add a detailed discussion on the advantages and disadvantages of different types of spectrometers in specific applications. Therefore, I recommend a minor revision.
1. What does it add to the subject area compared with other published material? The open database project proposed in the paper is a valuable contribution. As scientific research increasingly relies on interdisciplinary and inter-institutional collaboration, this database can serve as an essential platform for researchers to share and compare the performance of miniaturized spectrometers. It enhances data transparency and reproducibility, which will help to accelerate advancements in the field of integrated spectrometers.
2. Do you consider the topic original or relevant to the field? Does it address a specific gap in the field?
Yes, the topic is both original and highly relevant to the field. It addresses a specific gap in the existing research field: the lack of a comprehensive, open-source database for miniaturized spectrometers. However, I think the authors should provide more specific examples to show how the database can help researchers conduct analysis and comparison in actual research, and it is necessary to add a detailed discussion on the advantages and disadvantages of different types of spectrometers in specific applications.
In order to update the latest research progress, please consider citing the following papers:
Appl. Sci. 2024, 14(11), 4886. doi: 10.3390/app14114886
Opto-Electron Adv 7, 240099 (2024). doi: 10.29026/oea.2024.240099
Light Sci Appl 13, 278 (2024). doi: 10.1038/s41377-024-01638-4
Author Response
Dear Reviewer 1,
We thank you for your careful service in reading and commenting our manuscript.
We attach here the reply sent to the Editor and both you Referees.
Kind regards,

Reviewer 2 Report
Comments and Suggestions for Authors
The authors have written a comprehensive review of the field of miniaturized integrated spectrometers. The manuscript is well written and is giving a thorough introduction into the field of integrated spectrometers guiding the reader from the point of the different setups used to the actual achievements. This serves not only the people working in the field as a reference guide but also people new in the field. 
The implementation of the open-source database can be viewed as a milestone in the field that will spark new developments. 
I would therefore advise the editor to accept the manuscript for publication.
The paper should be reread to fix smaller errors in orthography and grammar.
Author Response
Dear Reviewer 2,
We thank you for your careful service in reading and commenting our manuscript.
We attach here the reply sent to the Editor and both you Referees.
Kind regards,
